

# Enhancing discovery in spatial data infrastructures using a search engine

Paolo Corti[1], Athanasios Tom Kralidis[2] and Benjamin Lewis[1]

[1] Center for Geographic Analysis, Harvard University, Cambridge, MA, USA
[2] Open Source Geospatial Foundation, Beaverton, OR, USA

## ABSTRACT

A spatial data infrastructure (SDI) is a framework of geospatial data, metadata, users and tools intended to provide an efficient and flexible way to use spatial information. One of the key software components of an SDI is the catalogue service which is needed to discover, query and manage the metadata. Catalogue services in an SDI are typically based on the Open Geospatial Consortium (OGC) Catalogue Service for the Web (CSW) standard which defines common interfaces for accessing the metadata information. A search engine is a software system capable of supporting fast and reliable search, which may use 'any means necessary' to get users to the resources they need quickly and efficiently. These techniques may include full text search, natural language processing, weighted results, fuzzy tolerance results, faceting, hit highlighting, recommendations and many others. In this paper we present an example of a search engine being added to an SDI to improve search against large collections of geospatial datasets. The Centre for Geographic Analysis (CGA) at Harvard University re-engineered the search component of its public domain SDI (Harvard WorldMap) which is based on the GeoNode platform. A search engine was added to the SDI stack to enhance the CSW catalogue discovery abilities. It is now possible to discover spatial datasets from metadata by using the standard search operations of the catalogue and to take advantage of the new abilities of the search engine, to return relevant and reliable content to SDI users.

## INTRODUCTION

A spatial data infrastructure (SDI) typically stores a large collection of metadata. While the Open Geospatial Consortium (OGC) recommends the use of the catalogue service for the web (CSW) standard to query these metadata, several important benefits can be obtained by pairing the CSW with a search engine platform within the SDI software stack.

### SDI, interoperability, and standards

An SDI is a framework of geospatial data, metadata, users and tools which provides a mechanism for publishing and updating geospatial information. An SDI provides the architectural underpinnings for the discovery, evaluation and use of geospatial information (*Nebert, 2004*; *Goodchild, Fu & Rich, 2007*; *Masó, Pons & Zabala, 2012*). SDIs are typically distributed in nature, and connected by disparate computing platforms and client/server design patterns.

Corresponding author
Paolo Corti, pcorti@gmail.com

A critical principle of an SDI is interoperability which can be defined as the ability of a system or components in a system to provide information sharing and inter-application cooperative process control through a mutual understanding of request and response mechanisms embodied in standards.

Standards (formal, de facto, community) provide three primary benefits for geospatial information: (a) portability: use and reuse of information and applications, (b) interoperability: multiple system information exchange and (c) maintainability: long term updating and effective use of a resource (*Groot & McLaughlin, 2000*). The OGC standards baseline has traditionally provided core standards definitions to major SDI activities. Along with other standards bodies (IETF, ISO, OASIS) and de facto/community efforts (Open Source Geospatial Foundation (OSGeo), etc.), OGC standards provide broadly accepted, mature specifications, profiles and best practices (*Kralidis, 2009*).

## Metadata search in an SDI and CSW

An SDI can contain a large number of geospatial datasets which may grow in number over time. The difficulty of finding a needle in such a haystack means a more effective metadata search mechanism is called for. Metadata is data about data, describe the content, quality, condition and other characteristics of data in order to ease the search and understanding of data (*Nogueras-Iso, Zarazaga-Soria & Muro-Medrano, 2005*). Metadata standards define a way to provide homogeneous information about the identification, the extent, the spatial and temporal aspects, the content, the spatial reference, the portrayal, distribution and other properties of digital geographic data and services (*ISO 19115-1: 2014, 2014*).

Ease of data discovery is a critical measure of the effectiveness of an SDI. The OGC CSW standard specify the interfaces and bindings, as well as a framework for defining the application profiles required to publish and access digital catalogues of metadata for geospatial data and services (*Open Geospatial Consortium, 2016*; *Nebert, Whiteside & Vretanos, 2005*; *Rajabifard, Kalantari & Binns, 2009*).

Based on the Dublin Core metadata information model, CSW supports broad interoperability around discovering geospatial data and services spatially, non-spatially, temporally, and via keywords or free text. CSW supports application profiles which allow for information communities to constrain and/or extend the CSW specification to satisfy specific discovery requirements and to realize tighter coupling and integration of geospatial data and services. The CSW ISO Application Profile is an example of a standard for geospatial data search which follows ISO geospatial metadata standards.

## CSW catalogue within the SDI architecture

In a typical SDI architecture the following components can be identified:

- *GIS clients*: desktop GIS tools or web based viewers.
- *Spatial data server*: returns geospatial data to map clients in a range of formats.
- *Cache data server*: returns cached tiles to map clients to improve performance.
- *Processing server*: responsible for the processing of the geospatial datasets.

- *Spatial repository*: a combination of a spatial database and file system, where the geospatial data is stored.
- *Catalogue server*: used by map clients to query the metadata of the spatial datasets to support discovery.

Desktop GIS clients generally access the SDI data directly from the spatial repository or the file system. When the user has appropriate permissions from these clients it is possible to perform advanced operations, which are generally faster than when performed over OGC web standards. Web based viewers access the SDI data served by the spatial data server using a number of OGC web standards over HTTP, typically WMS/WMTS/WMS-C when it is just needed to render the data, or WFS/WCS when it is needed to access to the native information respectively for vector or cover datasets. WFS-T can be used for editing vector datasets. Web viewers can run GIS SDI processes by using the WPS standard exposed by the processing server. All of these OGC standards can be used by desktop GIS clients as well. The spatial repository is generally a combination of a RDBMS with a spatial extension and the file system where are stored data which are not in the database.

The catalogue, based on the CSW standard, lets users to discover data and services in an SDI. CSW is a standard for exposing a catalogue of geospatial entities over the HTTP request/response cycle. In a SDI or portal CSW endpoints are provided by a CSW catalogue. Popular open source implementations of CSW catalogue include (but are not limited to) pycsw (http://pycsw.org/), GeoNetwork (https://geonetwork-opensource.org/), degree (https://www.deegree.org/) and Esri Geoportal Server (https://www.esri.com/en-us/arcgis/products/geoportal-server/overview).

A CSW catalogue implements a number of operations which are accessible via HTTP. Some of these operations are optional:

- *GetCapabilities* retrieves service metadata from the server.
- *DescribeRecord* allows a client to discover the data model of a specific catalogue information model.
- *GetRecords* searches for records using a series of criteria, which can be spatial, aspatial, logical, comparative.
- *GetRecordById* retrieves metadata for one record (layer) of the catalogue by its id.
- *GetDomain* (optional) retrieves runtime information about the range of values of a metadata record element or request parameter.
- *Harvest* (optional) creates or updates metadata with a request to the server to 'pull' metadata from some endpoint.
- *Transaction* (optional) creates or edits metadata with a request to the server.

## Need for a search engine within an SDI

Search workflow and user experience are a vital part of modern web-based applications. Numerous types of web application, such as Content Management Systems (CMS), wikis, data delivery frameworks, all can benefit from improved data discovery. Same applies

to SDI. Furthermore, in the Big Data era, more powerful mechanisms are needed to return relevant content to the users from very large collections of data (*Tsinaraki & Schade, 2016*).

In the last few years, content-driven platforms have delegated the task of search optimization to specific frameworks known as search engines. Rather than implementing a custom search logic, these platforms now often add a search engine in the stack to improve search. Apache Solr (http://lucene.apache.org/solr/) and Elasticsearch (https://www.elastic.co/), two popular open source search engine web platforms, both based on Apache Lucene (https://lucene.apache.org/), are commonly used in typical web application stacks to support complex search criteria, faceting, results highlighting, query spell-check, relevance tuning and more (*Smiley et al., 2015*). As for CMS's, SDI search can dramatically benefit from such platforms as well.

### How a search engine works

Typically the way a search engine works can be split into two distinct phases: indexing and searching. During the indexing phase, all of the documents (metadata, in the SDI context) that must be searched are scanned, and a list of search terms (an index) is built. For each search term, the index keeps track of the identifiers of the documents that contain the search term. During the searching phase only the index is looked at, and a list of the documents containing the given search term is quickly returned to the client. This indexed approach makes a search engine extremely fast in outputting results. On top of this, a search engine provides many other useful search related features, improving dramatically the experience of users.

### Improvements in an SDI with a search engine

There are numerous opportunities to enhance the functionality of the CSW specification and subsequent server implementations by specifying standard search engine functionality as enhancements to the standard. A search engine is extremely fast and scalable: by building and maintaining its indexed structure of the content, it can return results much faster and scale much better than a traditional CSW based on a relational database. While a CSW can search metadata with a full text approach, with a search engine it is possible to extend the full text search with features such as language stemming, thesaurus and synonyms, hit highlighting, wild-card matches and other 'fuzzy' matching techniques. Another key advantage is that search engines can provide relevancy scores for likely matches, allowing for much finer tuning of search results. CSW does not easily emit facets or facet counts as part of search results. Search engine facets however, can be based on numerous classification schemes, such as named geography, date and time extent, keywords, etc. and can be used to enable interactive feedback mechanisms which help users define and refine their searches effectively.

## BACKGROUND

Harvard WorldMap (http://worldmap.harvard.edu/) is an open source SDI and Geospatial Content Management System (GeoCMS) platform developed by the Centre for Geographic Analysis (CGA) to lower the barrier for scholars who wish to explore, visualize, edit and publish geospatial information (*Guan et al., 2012*). Registered users are

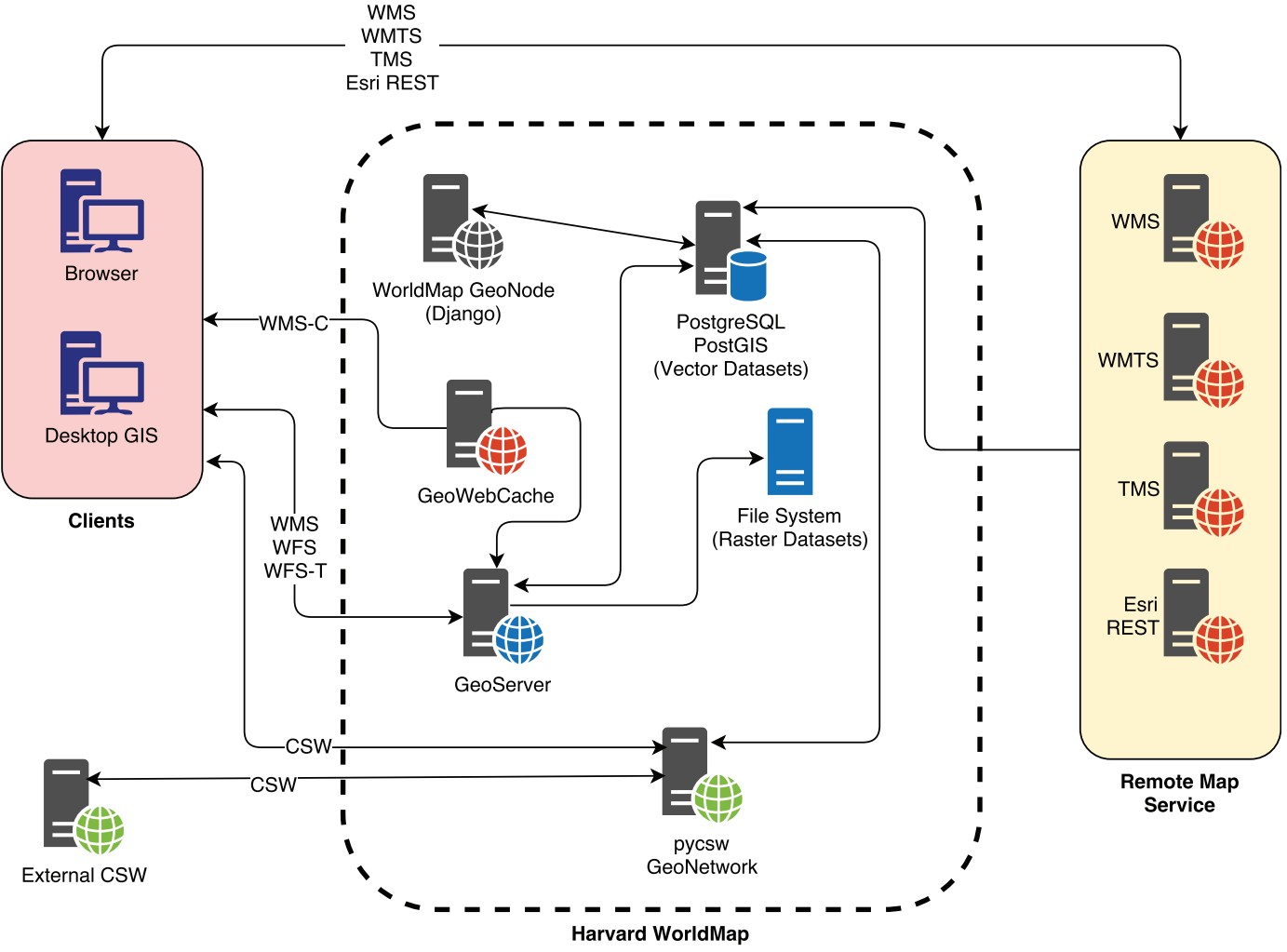

**Figure 1  The WorldMap SDI architecture.**

able to upload geospatial content, in the form of vector or raster datasets (layers), and combine them with existing layers to create maps. Existing layers can be layers uploaded by other users and layers provided by external map services.

WorldMap is a web application built on top of the GeoNode open source mapping platform (http://geonode.org/), and since 2010 has been used by more than 20,000 registered users to upload about 30,000 layers and to create some 5,000 web maps. Users can also access about 90,000 layers from remote map services based on OGC standards and Esri REST protocols.

WorldMap is based on the following components, all open source and designed around OGC standards (Fig. 1):

- A JavaScript web GIS client, GeoExplorer (http://suite.boundlessgeo.com/docs/latest/), based on OpenLayers (https://openlayers.org/) and ExtJS (https://www.sencha.com/products/extjs/).

- A spatial data server based on GeoServer (http://geoserver.org/).
- A cache data server based on GeoWebCache (http://geowebcache.org/).
- A spatial database implemented with PostgreSQL (https://www.postgresql.org/) and PostGIS (https://postgis.net/).
- A catalogue based on pycsw or GeoNetwork.
- A web application, developed with Django (https://www.djangoproject.com/), a Python web framework, which orchestrates all of the previous components.

WorldMap allows users to build maps using its internal catalogue of layers (local layers) combined with layers from external map services (remote layers), for a total of about 120,000 layers. WorldMap users can have trouble finding useful and reliable layers given the large number of them; a system was needed to enable fast, scalable search capable of returning the most reliable and useful layers within a large and heterogeneous collection.

## RESULTS AND DISCUSSION

In 2015 CGA started the design and development of Hypermap Registry (Hypermap) (https://github.com/cga-harvard/Hypermap-Registry) to improve search for WorldMap users. Hypermap is an application that manages OGC web services (such as WMS, WMTS, CSW Capabilities service metadata) as well as Esri REST endpoints. In addition it supports map service discovery (*Chen et al., 2011*), crawling (*Bone et al., 2016*; *Li, Yang & Yang, 2010*), harvesting and uptime statistics gathering for services and layers.

One of the main purposes of Hypermap is to bring enhanced search engine capabilities into an SDI architecture. As it can be seen from the following Fig. 2, search engine documents, based on a provided schema, must be kept in synchrony with layer metadata stored in the GeoNode RDBMS. Hypermap is responsible for ensuring that the WorldMap search engine, based on Apache Solr, and the WorldMap catalogue RDBMS, based on PostgreSQL, are kept in sync. For example, when a WorldMap user updates the metadata information for one layer from the WorldMap metadata editing interface, that information is updated in the WorldMap pycsw backend, which is based on the RDBMS. As soon as this happens, a synchronization task is sent from Hypermap to the task queue. The task will be processed by the task queue, and all of the metadata information for the layer will be synced to the corresponding search engine document.

Thanks to this synchronization mechanism, WorldMap users can search the existing layers metadata using a search engine rather than the OGC catalogue, enabling more flexible searches which filter on keywords, source, layer type, map extent and date range (*Corti & Lewis, 2017*). The results are returned by the search engine which returns a JSON response, and tabular in addition to spatial views (based on spatial facets) are returned to the browser (Fig. 2).

### WorldMap improvements with the search engine

By pairing the CSW catalogue with a search engine, the metadata search in the WorldMap SDI yields several major benefits.

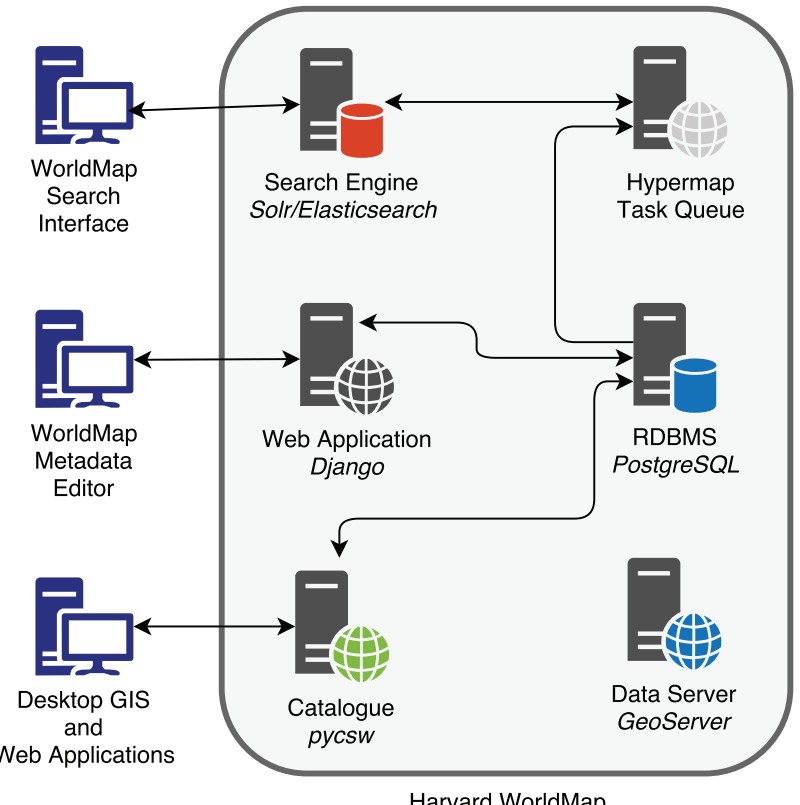

**Figure 2  Metadata RDBMS to search engine synchronization in Harvard WorldMap.**

### Fast results

By having the metadata content indexed in a search engine, metadata are returned very rapidly to the client. Because of its indexed documents nature, a search engine is much faster to return results when compared it with a relational database. Therefore, using a search engine in WorldMap search client makes things much faster than using a CSW catalogue based on a RDBMS.

### Scalability

From a software engineering perspective, search engines are highly scalable and replicable, thanks to their shardable architecture. Such systems are capable of providing interactive query access to collections of spatio-temporal objects containing billions of features (*Kakkar & Lewis, 2017*; *Kakkar et al., 2017*).

### Clean API

Query to the search engine API tends to be much simpler than XML queries to the CSW catalogue, specially when crafting advanced search requests (spatial, non-spatial, temporal, etc.). Same for output: JSON output from search engine API provides a more compact representation of search results enabling better performance and making the output more readable (Figs. 3 and 4).

```
Request:

<?xml version="1.0" ?>
<csw:GetRecords maxRecords="10" outputFormat="application/xml" outputSchema="http://www.opengis.net/cat/csw/2.0.2" resultType="results"
service="CSW" version="2.0.2" xmlns:csw="http://www.opengis.net/cat/csw/2.0.2" xmlns:ogc="http://www.opengis.net/ogc"
xmlns:xsi="http://www.w3.org/2001/XMLSchema-instance" xsi:schemaLocation="http://www.opengis.net/cat/csw/2.0.2
http://schemas.opengis.net/csw/2.0.2/CSW-discovery.xsd">
  <csw:Query typeNames="csw:Record">
    <csw:ElementSetName>full</csw:ElementSetName>
    <csw:Constraint version="1.1.0">
      <ogc:Filter>
        <ogc:PropertyIsLike escapeChar="\" singleChar="_" wildCard="%">
          <ogc:PropertyName>csw:AnyText</ogc:PropertyName>
          <ogc:Literal>libraries in boston</ogc:Literal>
        </ogc:PropertyIsLike>
      </ogc:Filter>
    </csw:Constraint>
  </csw:Query>
</csw:GetRecords>

Response:

<?xml version="1.0" encoding="UTF-8" standalone="no"?>
<!-- pycsw 2.1-dev-20161019 -->
<csw:GetRecordsResponse xmlns:csw="http://www.opengis.net/cat/csw/2.0.2" xmlns:dc="http://purl.org/dc/elements/1.1/"
xmlns:dct="http://purl.org/dc/terms/" xmlns:gmd="http://www.isotc211.org/2005/gmd" xmlns:gml="http://www.opengis.net/gml"
xmlns:ows="http://www.opengis.net/ows" xmlns:xs="http://www.w3.org/2001/XMLSchema" xmlns:xsi="http://www.w3.org/2001/XMLSchema-instance"
version="2.0.2" xsi:schemaLocation="http://www.opengis.net/cat/csw/2.0.2 http://schemas.opengis.net/csw/2.0.2/CSW-discovery.xsd">
  <csw:SearchStatus timestamp="2018-01-10T23:26:57Z"/>
  <csw:SearchResults nextRecord="0" numberOfRecordsMatched="1" numberOfRecordsReturned="1" recordSchema="http://www.opengis.net/cat/csw/2.0.2"
  elementSet="full">
    <csw:Record>
      <dc:identifier>b8482b92-c0b3-11e4-824f-22000aeecbbb</dc:identifier>
      <dc:title>Libraries in Boston</dc:title>
      <dct:alternative>geonode:alex_zku</dct:alternative>
      <dct:modified>2018-01-09T14:01:27Z</dct:modified>
      <dct:abstract>Showing the different libraries in the Boston area</dct:abstract>
      <dc:type>dataset</dc:type>
      <dc:format>Hypermap:WorldMap</dc:format>
      <dc:source>http://worldmap.harvard.edu/geoserver/geonode/geonode:alex_zku/wms?</dc:source>
      <dc:relation>2a96b71c-96b2-4432-b31f-219c45f3fc52</dc:relation>
      <dct:references scheme="Hypermap:WorldMap">http://worldmap.harvard.edu/geoserver/geonode/geonode:alex_zku/wms?</dct:references>
      <dct:references scheme="OGC:WMTS">http://localhost:8000/http://worldmap.harvard.edu/geoserver/geonode/geonode:alex_zku/wms?</dct:references>
      <ows:BoundingBox crs="http://www.opengis.net/def/crs/EPSG/0/4326" dimensions="2">
        <ows:LowerCorner>-1.0 -1.0</ows:LowerCorner>
        <ows:UpperCorner>0.0 0.0</ows:UpperCorner>
      </ows:BoundingBox>
    </csw:Record>
  </csw:SearchResults>
</csw:GetRecordsResponse>
```

**Figure 3** CSW request and response.

### Synonyms, text stemming

Crucially, search engines are good at handling the ambiguities of natural languages, thanks to stop words (words filtered out during the processing of text), stemming (ability to detect words derived from a common root), synonyms detection and controlled vocabularies such as thesauri and taxonomies. It is possible to do phrase searches and proximity searches (search for a phrase containing two different words separated by a specified number of words). Because of features like these, keyword queries using the Hypermap search engine endpoint typically returns more results than an equivalent

Request:

http://worldmap.harvard.edu/solr/hypermap/select?indent=on&q=_text_:%22libraries%20in%20boston%22&wt=json

Response:

```json
{
  "responseHeader":{
    "status":0,
    "QTime":1,
    "params":{
      "q":"_text_:\"libraries in boston\"",
      "indent":"on",
      "wt":"json"}},
  "response":{"numFound":1,"start":0,"docs":[
      {
        "abstract":"Showing the different libraries in the Boston area",
        "layer_username":"ambilaa",
        "reliability":97.96954314720813,
        "layer_originator":"worldmap.harvard.edu",
        "recent_reliability":100.0,
        "availability":"Online",
        "min_x":-1.0,
        "uuid":"b8482b92-c0b3-11e4-824f-22000aeecbbb",
        "title":"Libraries in Boston",
        "domain_name":"worldmap.harvard.edu",
        "id":190905,
        "is_valid":true,
        "location":"{\"layerInfoPage\": \"/registry/hypermap/layer/b8482b92-c0b3-11e4-824f-22000aeecbbb/\"}",
        "service_type":"Hypermap:WorldMap",
        "layer_category":["Place Locations"],
        "max_x":0.0,
        "max_y":0.0,
        "layer_datetype":"From Metadata",
        "layer_id":190905,
        "bbox":"ENVELOPE(-1.000000,0.000000,0.000000,-1.000000)",
        "last_status":true,
        "is_public":true,
        "min_y":-1.0,
        "layer_date":"2016-11-15T21:55:03Z",
        "name":"geonode:alex_zku",
        "url":"http://worldmap.harvard.edu/geoserver/geonode/geonode:alex_zku/wms?",
        "type":"Layer",
        "area":1.0,
        "tile_url":"/registry/hypermap/layer/190905/map/wmts/geonode:alex_zku/default_grid/{z}/{x}/{y}.png",
        "srs":["EPSG:4326",
          "EPSG:900913",
          "EPSG:3857"],
        "centroid_y":[-0.5],
        "centroid_x":[-0.5],
        "service_id":23,
        "_version_":1589125870789853184,
        "timestamp":"2018-01-09T14:35:28.508Z"}]
  }}
```

**Figure 4** **Search engine request and response.**

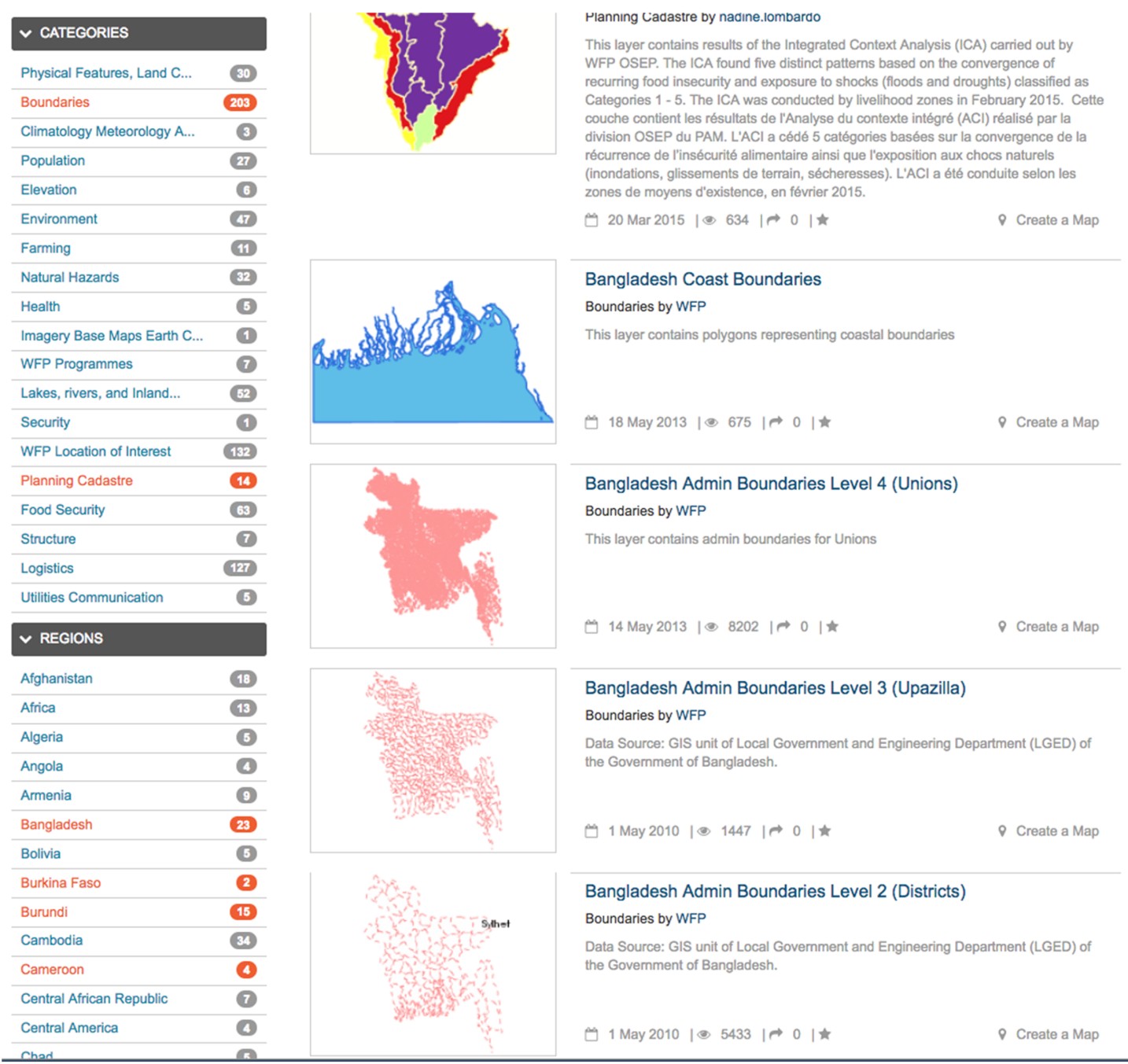

**Figure 5 Facets generate counts for metadata categories and geographic regions in a GeoCMS.**

query using the Hypermap CSW. For example doing a full text search for the keyword 'library' returns more results from the search engine because it includes variations and synonyms of the original term like 'libraries,' 'bibliotheca,' 'repository,' 'repositories' in the returned results.

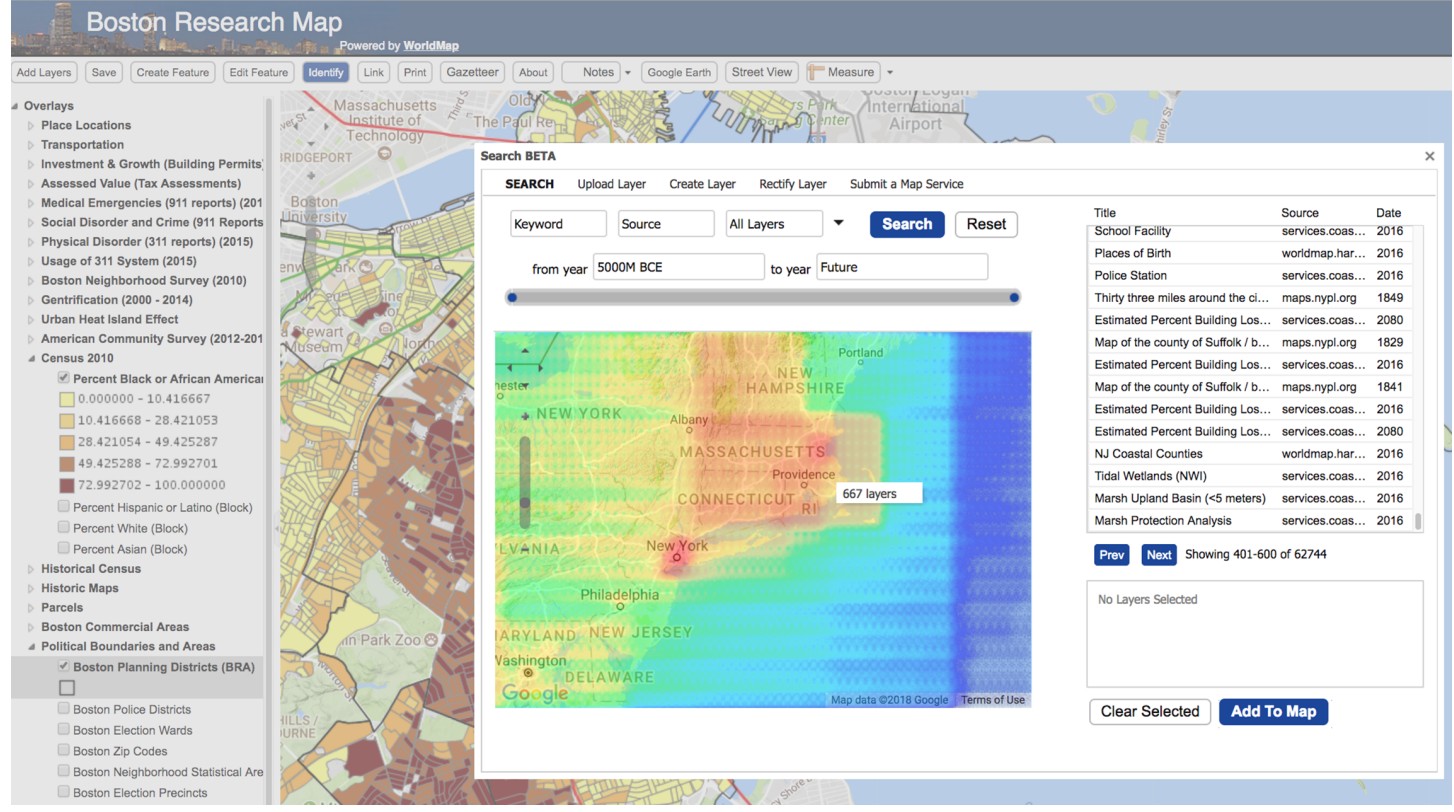

**Figure 6** Spatial faceting enables heatmaps showing the distribution of the SDI layers in the space.

### Relevancy

Results can be ranked, providing a way to return results to users with the more relevant ones closer to the top. This is very useful to detect the most significative metadata for a given query. Weights can be assigned by specifying boosts (weighted factors) for each field.

### Facets

Another important search engine feature useful for searching the WorldMap metadata catalogue is faceted search. Faceting is the arrangement of search results in categories based on indexed terms. This capability makes it possible, for example to provide an immediate indication of the number of times that common keywords are contained in different metadata documents. A typical use case is with metadata categories, keywords and regions. Thanks to facets, the user interface of an SDI catalogue can display counts for documents by category, keyword or region (Fig. 5).

Search engines can also support temporal and spatial faceting, two features that are extremely useful for browsing large collections of geospatial metadata. Temporal faceting can display the number of metadata documents by date range as a kind of histogram. Spatial faceting can provide a spatial surface representing the distribution of layers or features across an area of interest. In Fig. 6, a heatmap is generated by spatial faceting which shows the distribution of layers in the WorldMap SDI for a given geographic region (Fig. 6).

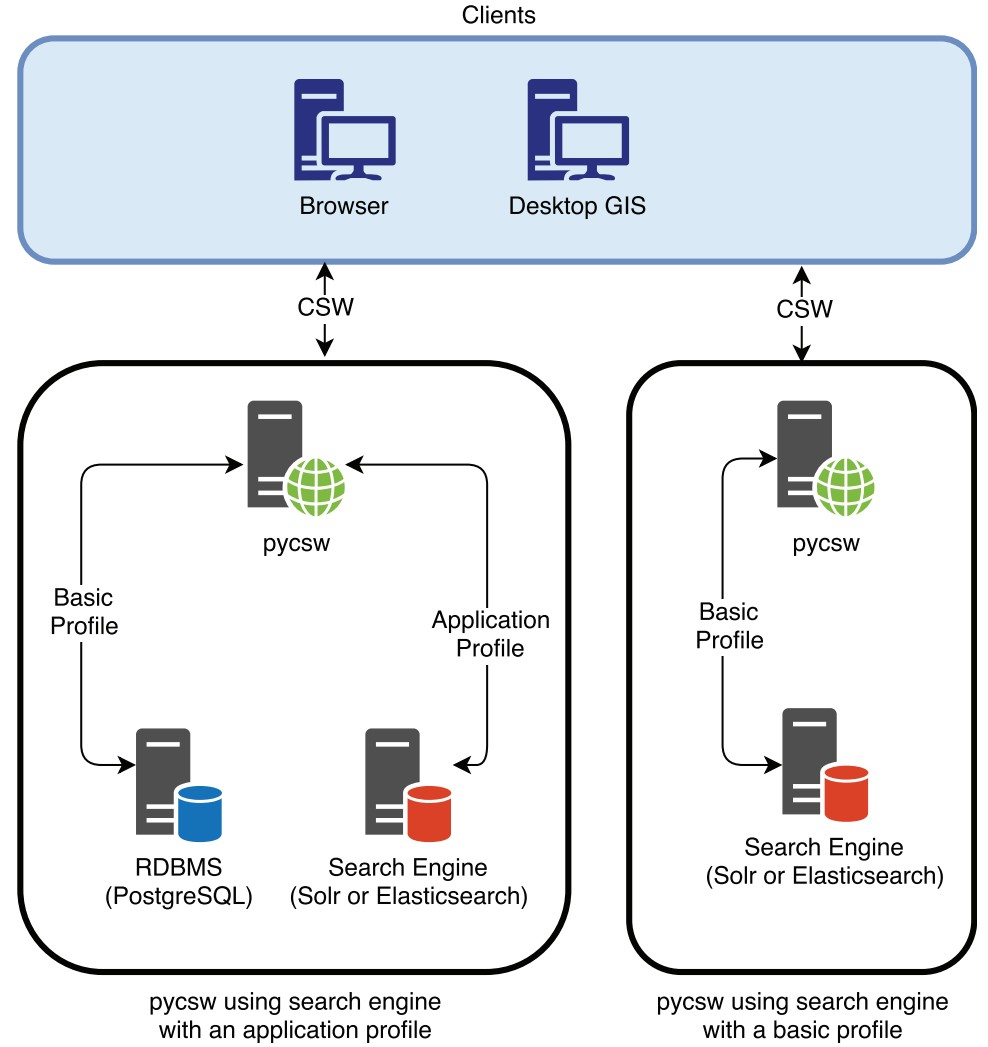

Clients

Browser        Desktop GIS

CSW                                CSW

pycsw                              pycsw

Basic
Profile                Application
                        Profile        Basic
                                        Profile

RDBMS            Search Engine
(PostgreSQL)    (Solr or Elasticsearch)        Search Engine
                                                (Solr or Elasticsearch)

pycsw using search engine              pycsw using search engine
with an application profile            with a basic profile

**Figure 7** **pycsw interaction with the search engine using an application profile and using a basic profile** (when pycsw will provide direct support for the search engine).

### Other features

In addition, it is possible to use regular expressions, wildcard search and fuzzy search to provide results for a given term and its common variations. It is also possible to support boolean queries: a user is able to search results using terms and boolean operators such as AND, OR, NOT and hit highlighting can provide immediate search term suggestions to the user searching a text string in metadata.

## CONCLUSION

While the CSW 3.0.0 standard provides improvements to address mass market search/discovery, the benefits of search engine implementations combined with broad interoperability of the CSW standard presents a great opportunity to enhance the CSW

standard. The authors hope that such an approach eventually becomes formalized as a CSW Application Profile or Best Practice in order to achieve maximum benefit and adoption in SDI activities. This will allow CSW implementations to make better use of search engine methodologies for improving the user search experience in SDI workflows.

In addition, pycsw is planning for dedicated Elasticsearch/Solr support as part of a future release to enable the use of search engines as backend stores to the CSW standard. This is a different approach from using an Application Profile or Best Practice, as it directly interacts with data in the search engine rather than in the RDBMS (Fig. 7).

The authors would like to share this work with the OGC CSW community in support of the evolution of the CSW specification. Given recent developments on the OGC WFS 3.0 standard (RESTful design patterns, JSON, etc.), there is an opportunity for CSW to evolve in alignment with WFS 3.0 in support of the principles of the W3C Spatial Data on the Web Best Practices (*Group, 2017*) in a manner similar to the work presented in this paper.

## ACKNOWLEDGEMENTS

The authors thank all the contributors to the Hypermap and GeoNode platform source code, particularly: Wayner Barrios, Matt Bertrand, Simone Dalmasso, Alessio Fabiani, Jorge Martínez Gómez, Wendy Guan, Jeffrey Johnson, Devika Kakkar, Jude Mwenda, Ariel Núñez, Luis Pallares, David Smiley, Charles Thao, Angelos Tzotsos, Mingda Zhang.

### Funding

This work is partially funded by the U.S. National Endowment for the Humanities, Digital Humanities Implementation Grant #HK5009113 and the U.S. National Science Foundation Industry-University Cooperative Research Centers Program (IUCRC) grant for the Spatiotemporal Thinking, Computing, and Applications Center (STC) #1338914, and by Harvard University. Grant administration was supported by Harvard's Institute for Quantitative Social Science. There was no additional external funding received for this study. The funders had no role in study design, data collection and analysis, decision to publish, or preparation of the manuscript.

### Grant Disclosures

The following grant information was disclosed by the authors:
U.S. National Endowment for the Humanities, Digital Humanities Implementation: #HK5009113.
U.S. National Science Foundation Industry-University Cooperative Research Centers Program (IUCRC).
Spatiotemporal Thinking, Computing, and Applications Center (STC): #1338914.
Harvard University.
Harvard's Institute for Quantitative Social Science.

## Competing Interests

The authors declare that they have no competing interests.

## Author Contributions

- Paolo Corti conceived and designed the experiments, performed the experiments, analyzed the data, contributed reagents/materials/analysis tools, prepared figures and/or tables, performed the computation work, authored or reviewed drafts of the paper, approved the final draft.
- Athanasios Tom Kralidis performed the experiments, analyzed the data, contributed reagents/materials/analysis tools, performed the computation work, authored or reviewed drafts of the paper, approved the final draft.
- Benjamin Lewis performed the experiments, analyzed the data, contributed reagents/materials/analysis tools, performed the computation work, authored or reviewed drafts of the paper, approved the final draft.

## Data Availability

Hypermap Registry: https://github.com/cga-harvard/Hypermap-Registry

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
