# Peer review of "Enhancing discovery in spatial data infrastructures using a search engine"

_PeerJ Computer Science, doi:10.7717/peerj-cs.152_

## Round 0.1 · original submission · Minor Revisions

On the basis of the of the two reviewers' comments, the paper is worth being published on PeerJ Computer Science. Reviewer 2 suggests that the paper is ready to be accepted.

However, Reviewer 1 has several points to be discussed which I think would help to improve even more the already good paper you have submitted.

It would be nice also to have, some more details on performances, possible issues, and planned/preferable development scenarios.

Therefore I suggest to revise the manuscript following the reviewers comments below.

·

Basic reporting

The article is clearly written but has some unclearness related to the scope and structure of the content.

Experimental design

The proposed solution seems to be relevant and addresses a significant criticality of the distributed SDIs, but the structure of the article makes its main aim unclear.

Validity of the findings

The architecture and solutions described are solid and of importance in the field of SDIs but conclusions suffer from the not clear focus of the article.

Additional comments

The proposed solution seems to be relevant and addresses a significant criticality of the distributed SDIs. However, the focus / main objective of the article is not always clear. Is the main objective to describe a new conceptual approach/model and architecture for the use of search engines in order to support the discovery?; or is the main focus on the hhypermap software?; or again, about its use within WorldMap?

Probably the authors should immediately explain these aspects and use them as a fil rouge to better organize the subsequent parts of the paper (method, results, discussion) that sometimes do not seem clearly organized.

Main issues:
- it may be useful (but this also depends on the focus) to better separate the general architecture part (SDI + search engine) from specific solutions/implementations made through GeoNode, HHypermal etc. This could help the general understanding of the problem and give the paper a wider relevance
- again with reference to these aspects, the description made of the architecture (in particular for Fig. 2) is minimal and could be developed better
- if the focus of the project wants to be Hhypermap we could better evaluate the fact that it is released as open source, which is only briefly mentioned in the abstract of the article, adding also direct references to the repository. A brief explanation (or clear reference to documentation) on how to install an use the software in this case would be really beneficial.

Reviewer 2 ·

Basic reporting

Good and precise use of the English language applies.

Introduction and Background demonstrate the authors full grasp of both of the geospatial and library science aspects of the discussed topic, including applicable state-of-the art standards and software projects.

The document structure conforms to the PeerJ standards. Figures are relevant, well drawn and correctly labeled. Links to the project web pages of the described software tools (both for results and used sources) are provided, which point to the respective software repositories.

Experimental design

The research is within the Scope of the journal.

The research question is well defined.

The descriptions of the HyperMap and WorldMap software stacks, the architectural views, description of functionality and examples of input/output (Figure 3) are very concise.

The conclusion, that search engine technology should be embraced in future editions of the CSW OGC standard is sound and would would make a lot of sense.

Validity of the findings

The results are valid and a step forward in the advancement of Science, as the impact on the improvement of the OGC CSW standard.

The developed research code is available for review and reuse from github repositories.


It would be interesting to learn in future reports about the topic (which are highly encouraged) more about the speed improvement (line 165: "about 20/30 times faster" to infer benefits of scaling effects for application scenarios which are currently still prohibitive due to technical limitations (CSW standard definition).

---

## Round 0.2 · accepted · Accept

I'm pleased to see that reviewer's comments and suggestions have been implemented meticulously, improving the quality of the article which now I recommend for publication in the PeerJ Computer Science journal.

I'm glad that the you have targeted the PeerJ Computer Science journal as the spatial data discovery subject is becoming of great importance in the field of spatial/geographic data information system.

I encourage you to look at PeerJ Computer Science for the publication of your future studies in this field.

·

Basic reporting

no comment

Experimental design

no comment

Validity of the findings

no comment

Additional comments

The authors modified the manuscript accordingly to the comments provided, clarifying the highlighted issues, improving the structure and clarifying the objectives of the study.
I recommend this article for publication.